# The Psychosocial Risks and Impacts in the Workplace Assessment Tool: Construction and Psychometric Evaluation

**DOI:** 10.3390/bs13020104

**Published:** 2023-01-27

**Authors:** Petros L. Roussos

**Affiliations:** Department of Psychology, National and Kapodistrian University of Athens, 10679 Athens, Greece; roussosp@psych.uoa.gr

**Keywords:** psychosocial risks, psychometric tool, PRIWA, occupational health

## Abstract

Psychosocial risks constitute one of the major contemporary challenges for occupational health and safety. As early identification is the first step towards psychosocial risk management, the psychometric tool presented in the paper has been constructed in order to measure psychosocial risks as well as their impacts. The Psychosocial Risks and Impacts in the Workplace Assessment Tool (PRIWA) has been developed in Greek during the early years of the economic crisis. The paper presents the tool and the studies that were conducted to evaluate its psychometric characteristics. Six large samples of employees from many different Greek companies were administered the PRIWA and other tools. The results of the exploratory factor analysis demonstrated a seven-factor structure of the PRIWA, which was later confirmed by confirmatory factor analysis. Analyses were also performed to test internal consistency, item-to-scale homogeneity, and concurrent validity of the PRIWA. The results indicated that PRIWA is a reliable and valid psychometric tool, which gives its users the opportunity to conduct research, develop prevention plans, and/or design customized interventions.

## 1. Introduction

Over the last 15 years, work environments have evolved as a result of factors such as the automatization of work practices through new technology, workload and work intensity, growth in the service sector, new employment trends, etc. This shift in the world of work introduced situation risks that affect the health and safety of employees [1]. Those changes—besides physical, biological, and chemical risks—have led to emerging psychosocial risks. Psychosocial risks, work-related stress, violence, and harassment are recognized as major challenges to occupational health and safety [2]. Societal factors, including economic recession, encumber/aggravate psychosocial risks at the workplace [3]. Psychosocial risks are associated with serious economic implications for all types of enterprises, irrespective of size and sector [4].

According to the International Labour Organization [5], psychosocial hazards are related to factors that interact with each other. Those factors include environmental and organizational conditions, such as job context and work organization and management as well as employees’ competencies and needs. Psychosocial hazards are relevant to imbalances in the psychosocial arena and refer to those interactions that prove to have a hazardous influence over employees’ health through their perceptions and experience. Put simply, the term ‘psychosocial hazards’ has been defined in relation to the interaction between psychological and social factors [6]. Psychosocial hazards refer to those aspects of work design, the organization and management of work, and their social and environmental contexts, which have the potential to cause psychological, social, or physical harm [7].

There is a consensus regarding the nature of psychosocial hazards [8]. Psychosocial hazards include factors such as organizational culture and function, role in the organization, workload and work pace, interpersonal relationships at work, etc. However, the changes that are constantly made in the workplace give rise to new risks that have not yet been identified and, therefore, are not presented in scientific publications. Several changing work patterns have been identified, such as downsizing, outsourcing, increase in the use of information and communication technology (ICT) in the workplace, self-regulated work, additional demands for workers’ flexibility related to the number and function of skills, shift work, and unsocial hours, as factors that lead to new forms of psychosocial risks [9].

Work experiences can have a tremendous impact on an employee’s psychological state of health [10]. Research has shown that the impact of new working patterns and the risks that accompany them are apparent in employees’ health and especially in their stress levels. Continuous exposure to psychosocial risk factors may result in work-related stress for employees, affecting their efficiency in performing tasks [3]. Occupational stress has received great attention as it has become almost synonymous with work life. Nearly one in three workers in Europe, more than 40 million people, report that they are affected by stress in the workplace [9]. In the 15 Member States of the pre-2004 EU, the cost of stress at work and related mental health problems was estimated to be on average between 3% and 4% of the gross national product, amounting to €265 billion annually [11]. Poorly managed organizational change can result in increased stress [12,13] and reduced job satisfaction and organizational commitment [12].

Furthermore, evidence shows that stress is related to poorer performance, higher absenteeism, and an increase in accident rates [6]. In the UK, stress, anxiety, and depression account for 13.8 million days lost, or 46% of all reported illnesses, making it the single largest cause of absence related to work-related illness. Since 2004, work-related stress, depression, or anxiety is, for each year, the most reported complaint. It is stated that employers underestimate the extent to which employees are experiencing forms of mental ill health such as stress, depression, and anxiety as well as the detrimental effects that mental ill health may be having on their business [14,15]. Absence due to psychological problems is as common as due to musculoskeletal disorders [6,16]. 

Similarly, exposure to psychosocial risks can lead to presenteeism. Research on presenteeism showed that its costs are 1.8 times as important as absenteeism. In comparison to absenteeism, presenteeism has a larger effect, and mental ill health is more likely to be manifested with presenteeism rather than absenteeism [17]. Violence and harassment are also recognized as major challenges to occupational health and safety [2].

According to the European Working Conditions Surveys and the scientific literature, some occupations are particularly at risk from different forms of violence. In 2005, threats of physical violence were reported by workers employed in sectors such as education and health (14.6%) and transport and communication (9.8%). Harassment was mostly reported in certain occupational sectors such as hotels and restaurants (reported by 8.6% of workers), whilst the highest rate of unwanted sexual attention was reported by 3.9% of workers in hotels and restaurants [6]. Additionally, there is emerging evidence that there is a link between workplace characteristics and risky patterns of alcohol consumption [18] and the increasing precariousness of work [19]. 

Psychosocial risk management has been approached as synonymous with best business practices. As such, best practices in relation to psychosocial risk management essentially reflects best practices in terms of organizational management, leadership and development, social responsibility, and the promotion of quality of working life and good work. Positive and adaptive mental health states (e.g., happiness and engagement) have been recognized as important for promoting physical health and psychological wellbeing. For instance, high involvement has been linked with desirable outcomes such as job satisfaction and organizational commitment.

Psychosocial risks can and should be included in general strategic methods in risk assessment carried out by employers. According to Directive 89/391/EEC, psychosocial risk management is also among employers’ responsibilities to prevent, manage, and establish health and safety methods [20].

Assessing psychosocial issues is considered the first step in the psychosocial risk management process. Psychosocial risks can be recorded/documented by using a variety of measures such as examining archival data, observations during work, focus groups, conducting employee interviews, and administering self-reported surveys. The most widely used measures are self-reported surveys for identifying the risk factors linked to job stress and other negative outcomes in the workplace.

Nearly half of the questionnaires and most of the observational questionnaires on psychosocial risks are used by national institutions or in national initiatives. Some questionnaires were originally developed for the specific purposes of an institutional initiative such as the Copenhagen Psychosocial Questionnaire (COPSOQ) in Denmark [21]. Such instruments form part of broader public prevention campaigns. For instance, in the UK, the Health and Safety Executive (HSE) Indicator tool constitutes part of a broader institutional approach administered under the specific requirements of the UK Management of Health and Safety at Work Regulations [22]. Similarly, in the Netherlands, instruments such as Welzijn Bij of Arbeid (WEBA) and Vragenlijst Beleving En Beoordeling Van De Arbeid (VBBA) have been administered for the Occupational Health and Safety Services by Work and Organisational Experts within the framework of a public prevention campaign. Some of the instruments that have been developed constitute initiatives for work stress control such as the checklist on work-related stress that has been administered in 2012 in certain occupational sectors as part of the European campaign on psychosocial risks at work under the initiative of the Committee of Senior Labour Inspectors.

Based on the above literature and research results, it was decided to construct a tool that would combine the measurement of psychosocial risks in the workplace as well as the outcomes of those risks in employees. As shown above, although numerous tools have been developed measuring a combination of psychosocial risks, the need for developing new questionnaires in the field of psychosocial risks can be attributed to multiple factors driven, among others, by socioeconomic transformations including social changes. The instrument was developed based on the following theoretical considerations and basic principles: (a) it should be theory-based, covering potential work stressors as well as resources such as support, feedback, commitment, and good physical and mental health (it follows the European framework for psychosocial risk management, which is rather broad and aims at accommodating differences in approach and culture across the European countries) [3,8], (b) it should consist of dimensions related to different levels of analysis (organization, person–work interface, and individual), (c) it should include dimensions related to work tasks, the organization of work, interpersonal relations at work, cooperation, and leadership, (d) it should be comprehensive, and (e) it should be applicable in all sectors of the labor market.

Up until now and according to relevant research on instruments on psychosocial issues [23], most of the tools exclusively measure a variety of the identified psychosocial risks. For instance, the General Nordic Questionnaire for Psychological and Social Factors at Work (QPS NORDIC) measures variables such as job demands, role expectation, communication, and work culture. The QPS NORDIC measures those factors as potential determinants of motivation, health, and wellbeing without actually measuring the effects of those risks [24]. Similarly, the Nova Weba Questionnaire is concentrated on measuring stress-related risks, i.e., control requirements/job demands, control options, etc. Other instruments measure more specific areas of psychosocial risks. The Job Diagnostic Survey (JDS), for instance, is designed to measure variables that are used to examine how workers respond to job design, such as autonomy, skill variety, task significance, etc. [23].

The present study was designed to develop a new instrument for the assessment of workplace psychosocial risks and their impacts in the Greek language. The psychometric tool presented in this paper can be applied as a survey feedback instrument in psychosocial risk management and organizational development. It has been constructed for assessing employees’ perceptions of psychosocial conditions in the workplace and their reactions to stressors in terms of psychological wellbeing (stress and anxiety, alcohol consumption, etc.) and organizational commitment (satisfaction and engagement and presenteeism) with the following goals: provide the basis for psychosocial risk management and organizational development, design customized interventions based on specific results, document the changes regarding the working conditions, and conduct research on the relationship between the work situation, health, wellbeing, and productivity.

## 2. Materials and Methods

### 2.1. Instrument Development

A pool of positive and negative statements about individual and organizational risk factors was developed using items written for the instrument following a number of focus group meetings with fellow researchers, HR professionals, and occupational health professionals. In this way, 230 items were developed covering work/life balance; physical (physical complaints and physical health), cognitive (mental exhaustion and feelings of insufficiency), and emotional (depression, anxiety, and job satisfaction) reactions to the demanding work situation; negative acts in the form of being a victim of physical, verbal, or sexual harassment or mobbing; absenteeism; issues related to organizational culture, climate, and function; organizational change (positive or negative); autonomy and control over work; job demands (workload, work pace, and poor job fit) and security; interpersonal relationships at work; and the employee’s role in the organization (role expectations, lack of respect and lack of appropriate rewards and recognition). Subsequently, Sample 1 used a 5-point Likert response scale that ranged from 1 “strongly disagree” to 5 “strongly agree” to indicate their level of agreement or disagreement with each of these items.

Items were screened for their tendency to elicit extreme responses, with items being discarded if they produced mean responses of more than 4.5 or less than 1.5 on the 5-point Likert-type scale employed; twenty-three items were discarded on these grounds. Therefore, principal components analysis was performed on the remaining 207 items (see Section 3).

#### 2.1.1. Copenhagen Psychosocial Questionnaire

The Copenhagen Psychosocial Questionnaire (COPSOQ II) was developed in 1997 by the Danish National Research Centre for the Working Environment as a standardized questionnaire covering a broad range of psychosocial factors [25]. It includes three different-length versions: a long version for research and a medium-length and a short version for risk assessment purposes in the workplace. Parts of the medium-length and the short version were used in the present study: specifically, five scales of the short version (10 items measuring predictability, quality of leadership, social support, feedback at work, and sense of community) were administered to Sample 4, and twelve scales of the medium-length version (47 items measuring general health, mental health, vitality, behavioral stress, somatic stress, cognitive Stress, quantitative demands, cognitive demands, emotional demands, sensorial demands, demands for hiding emotions, and insecurity at work) were administered to Sample 3. The internal consistency of the scales was high, with Cronbach’s alpha values above 0.7 in 19 of the 24 dimensions, the lowest value being 0.59 for the dimension of predictability. The translation of its English version into Greek was performed by two bilingual experts in work and counseling psychology. Back-translation was used as a means of quality control.

#### 2.1.2. Utrecht Work Engagement Scale

The Utrecht Work Engagement Scale (UWES) includes three constituting dimensions of work engagement: vigor, dedication, and absorption. The shortened version of the UWES (UWES-9) was used in the present study (it was administered to Sample 5), which contains nine items [26]. The factorial validity of the tool was demonstrated using confirmatory factor analyses, and the three scale scores showed good internal consistency (Cronbach’s alpha of the three scales varied across countries between 0.60 and 0.88) and test–retest reliability (stability coefficients for the three scales across countries varied between 0.56 and 0.73).

#### 2.1.3. The Stanford Presenteeism Scale

This is the 6-item instrument (SPS-6) developed by Koopman et al. [27], which evaluates the impact of health problems on individual work performance and overall perceived productivity for knowledge-based activity. This is a tool used for monitoring and measuring interventions and improvement of employee health status and productivity. The instrument is not norm-referenced or standardized, but higher scores are associated with higher presenteeism or a greater perceived ability to concentrate on and accomplish work despite health problems. It has good psychometric properties (Cronbach’s α = 0.80 showing high internal consistency; concurrent, criterion, and discriminant validity measures were used to test the validity of the instrument with very good results) and has emerged as a tool utilized in employee health and wellness interventions addressing changes in presenteeism for medical residents and employees as well as the impact of chronic illness and mental health problems on presenteeism [28].

### 2.2. Participants

Six different convenience samples were used in this study to develop (Sample 1) and investigate the factor structure (Sample 2) and later evaluate the psychometric properties (internal consistency, test–retest reliability, and validity) of the PRIWA (Samples 3, 4, 5, and 6). The research work reported here began in 2013 with the administration of the initial instrument to Sample 1 and ended in 2019.

Sample 1 consisted of 1032 participants (59.6% females and 38.6% males) ranging in age from 19 to 66 years (mean = 34.8, SD = 9.7 years) and was administered the 230 initial items that were developed for the PRIWA. They were an opportunity sample from more than fifty companies from various areas in Greece.

The second sample received only the 68-item PRIWA and consisted of 1309 participants (59.3% females and 40% males; mean age = 36.1 (9.8) years) who were employees from many different Greek companies.

The respondents of Sample 3 were a subgroup of Sample 2; they received the 68-item PRIWA along with the Copenhagen Psychosocial Questionnaire (COPSOQ II). This opportunity sample consisted of 494 participants (55.7% females and 42.3% males; mean age = 39.2 (9.7) years) who were employees from six major Greek companies.

Sample 4 included 402 participants (10% females and 90% males) ranging in age from 31 to 60 years (mean = 43.5, SD = 5.9 years) from a public transport bus company. The response rate approached 15% of the company’s workforce, and participants were administered the 68-item PRIWA along with five scales of the short version of the Copenhagen Psychosocial Questionnaire (predictability, quality of leadership, social support, feedback at work, and sense of community).

The fifth sample received the 68-item PRIWA along with the Utrecht Work Engagement Scale (UWES). This was also a convenience sample and consisted of 283 participants who were working in more than 15 public and private companies (65.1% females and 34.9% males; mean age = 40.1 (10.3) years). Two to three weeks after initial testing, the participants in Sample 5 retook the PRIWA.

Finally, sample six received the 68-item PRIWA along with the Stanford Presenteeism Scale (SPS-6) [27]. This was also a convenience sample of 148 workers from various companies (58.4% females and 40.9% males; mean age = 37.9 (11.4) years).

Table 1 presents the sociodemographic characteristics of the six samples.

Only the participants without missing values on the critical measures in the current study were selected for analyses.

### 2.3. Procedure

Most of the participants were administered paper-and-pencil questionnaires. However, approximately half of the data were collected using a questionnaire administered online through SurveyMonkey software.

Participants in Sample 1 only responded to the questionnaire with the initial 230 items, and the data collected from these administrations were used for item analysis only. Participants in Sample 3 were administered only the 68-item PRIWA to analyze the construct validity of the tool. Finally, participants in Samples 2, 4, 5, and 6 were administered the 68-item PRIWA along with scales from the COPSOQ (Samples 2 and 4), the UES (Sample 5), and the SPS-6 (Sample 6) to test the convergent validity of the tool, and Sample 5 retook PRIWA to measure its test–retest reliability.

Inclusion criteria were employees aged 18 years old and above who were proficient in the Greek language and agreed to participate in the study. Exclusion criteria were employees who did not give their consent and were not proficient in the Greek language.

Demographic information for these samples included sex, age, level of education, marital status, number of kids, employment sector, occupational status, and type of job contract. In all studies, participants remained anonymous, and no personal data were revealed to their companies.

## 3. Results

### 3.1. Exploratory Factor Analysis

Principal components analysis (PCA) was conducted on the 207 items with oblique rotation (direct oblimin). The Kaiser–Leyer–Olkin measure verified the sampling adequacy for the analysis; KMO = 0.93, which is considered excellent [29]. Bartlett’s test of sphericity, χ^2^(2278) = 263,120.38, *p* < 0.001, indicated that correlations between items were sufficiently large for PCA. An initial analysis was run to obtain eigenvalues for each component in the data. Forty-eight components had eigenvalues over Kaiser’s criterion of 1 and in combination explained 64.7% of the variance. The scree plot was rather ambiguous and showed inflections that would justify at least three different solutions with the number of components ranging between four and nine. The seven-factor solution seemed to be the most meaningful (the seven factors covered all three indicators for psychological risk assessment as described by the European framework PRIMA, namely organizational factors, work-related factors, and outcomes) [3,8] and was finally preferred. During several steps, a total of 139 items were eliminated because they did not contribute to a simple factor structure and failed to meet the minimum criterion of having a primary factor loading of 0.4 or above and no crossloading of 0.3 or above.

Table 2 shows the factor loadings after direct oblimin (oblique) rotation. Usually, only the pattern matrix is interpreted because it is simpler [30]; however, the structure matrix is useful in testing a potential drawback of oblique rotation: interpretation of the factors may be more difficult due to a suppression effect [31]. Therefore, both sets of standardized coefficients, pattern, and structure are reported as a double-check [32]. The two signs of suppression are when (a) the sizes and (b) the signs of the pattern coefficient and structure coefficient for the same indicator are different. None of these problems appeared, and the two matrices are very similar. The seven-factor solution explained 47.5% of the variance. The items that cluster on the same components suggest that component 1 represents organizational culture (the component explains 20.1% of the variance), component 2 health and wellbeing (7.5%), component 3 dysfunctional behaviors (5.8%), component 4 job satisfaction and engagement (3.5%), component 5 work demands (3.0%), component 6 job insecurity (4.8%), and component 7 presenteeism (2.7%).

Table 3 displays Pearson’s correlation coefficients between the scores of the 1051 participants on the seven factors of the 68-item PRIWA.

The results indicate that zero to low positive correlations exist between the total scores of the seven components, suggesting independence between the components. The only moderate coefficient was found between the total scores of the first and fourth components of the PRIWA (r = 0.57, *p* < 0.001), which are related conceptually.

### 3.2. Confirmatory Factor Analysis

In order to test the factorial validity of the scores from the PRIWA, confirmatory factor analyses (CFAs) were performed. Table 4 displays the results of the three CFAs, which were conducted using AMOS 21 [33] on the data collected from Sample 2. Three models were tested: Model 1 was a single-factor model (with 68 items), Model 2 was the seven-factor model revealed by the PCA, and Model 3 was the seven-factor model with covariances among errors. Eleven such error covariances were estimated, strictly within factors, thus not allowing any crossloadings involvement in our modeling. The criteria used for the assessment evaluation of the overall goodness of fit for the measurement model were (a) the chi-square/degrees of freedom ratio, (b) the robust comparative fit index (CFI), (c) the Tucker–Lewis index (TLI), (d) the standardized root mean square residual (SRMR), and (e) the root mean square error of approximation (RMSEA) with 90% confidence intervals (CI90). To compare the three models, the Akaike information criterion (AIC) was used with values’ reduction indicating model improvement. The chi-square/degrees of freedom ratio should have a value less than 2.0 or 3.0 to indicate a good fit, whereas CFI and TLI values > 0.95 are generally accepted as indicating a good fit, and for the RMSEA, an adequately fitting model has a value below 0.06 with upper 90% CI below 0.08 [34]. The chi-square result was statistically significant in all cases (*p* < 0.001), thereby suggesting that the fit of the data to the hypothesized models was not entirely adequate. However, χ^2^ is greatly affected by sample size, and findings of well-fitting hypothesized models have proven to be unrealistic in most SEM research [35]. The chi-square/degrees of freedom ratio was less than 3.0 in Model 3 (20.61), which is considered a satisfactory value by some researchers [36]; different researchers [35] argue that values above 2.0 indicate an inadequate fit. The values of the CFI and the TLI were not satisfactory for Models 1 and 2, but they were acceptable for Model 3 [37]. The SRMR and the RMSEA satisfied the criteria in all cases except that of Model 1.

As shown in Table 4, the third model provided improved fit indices, indicating a good fit between the 68-item seven-factor model and the observed data, and exhibited a lower AIC value than the other two models. In addition to the above analyses, Model 3 was also tested with the data collected from Sample 4. The results (χ^2^/df=10.93, CFI = 0.92, TLI = 0.91, SRMR = 0.053, RMSEA = 0.048 (CI 90%, 0.46−0.50)) confirmed the good fit between the model and the observed data [37].

### 3.3. Internal Consistency and Test–Retest Reliability

To evaluate the internal consistency of the PRIWA, the Cronbach’s alpha index was calculated with an acceptable standard of >0.70 [38]. Cronbach’s α values ranged from 0.74 to 0.92 for the seven factors of the PRIWA (Table 5).

The test–retest data collected from Sample 5 yielded statistically significant, positive correlations (Pearson’s correlation coefficients ranged between r = 0.78 and r = 0.92; all significant at *p* < 00.001).

### 3.4. Validating Measures

The concurrent validity of the PRIWA was measured by calculating Pearson’s correlation coefficients between the PRIWA scales and the scores of three other tools using four different samples (see Table 6). Specifically, Sample 4 was used to correlate the score of five scales of the Copenhagen Psychosocial Questionnaire (predictability, quality of leadership, social support, feedback at work, and sense of community) with the organizational culture construct of PRIWA (Factor 1). Using Sample 3 data, the COPSOQ II score calculated from the health scales (general health, mental health, and vitality) was correlated with the PRIWA health and wellbeing construct (Factor 2), the COPSOQ II score calculated from the stress scales (behavioral, somatic, and cognitive stress) was correlated with the dysfunctional behaviors PRIWA construct (Factor 3), the scores of the five scales measuring demands (quantitative, cognitive, emotional, and sensorial demands and demands for hiding emotions) were correlated with the PRIWA score of work demands (Factor 5), and the COPSOQ II score of insecurity at work scale was correlated with the PRIWA job insecurity score (Factor 6). Sample 5 data were used to correlate the three UWES scores on vigor, dedication, and absorption with the job satisfaction and engagement score of PRIWA (Factor 4). Finally, the SPS-6 score of Sample 6 was correlated with the score of the last PRIWA construct (presenteeism). The correlations were all moderate to strong, ranging from r = 0.60 to r = 0.83 (all statistically significant at the 0.001 level).

## 4. Discussion

PRIWA is the first instrument that has been constructed under the socioeconomic conditions that the economic crisis has imposed in Greece. PRIWA takes into account the continuous changes that have taken place in organizations including restructuring, layoff, and merges and aims to assess the range of impacts of psychosocial risks that may rise under such conditions.

As already shown, there are instruments that combine psychosocial risk variables as well as the outcomes of those variables, such as the COPSOQ. However, in comparison to other instruments on psychosocial risks, PRIWA assesses a variety of variables/factors that are designed to measure outcomes of psychosocial issues that still have not been included in the existing instruments. For instance, PRIWA measures satisfaction and engagement, addictions, signs of depression, stress, anxiety, etc. PRIWA also constitutes the first instrument on psychosocial risks that assesses presenteeism, a variable that has received great attention recently due to its consequences on an organization’s productivity and profit.

PRIWA is an initiative that derives from the private domain directed towards organizations in the public and private sector and focuses on prevention, risk identification, and early intervention. We have been using the instrument for a few years now in a considerable number of workplaces all over Greece, and PRIWA has become a popular and useful tool for both research and preventive practice in the workplace. The identification of risks is vital in the development of effective preventive and early intervention services. PRIWA constitutes a tool that focuses on the diagnosis of the implications of psychosocial risks so as to provide employers with customized interventions. Those interventions will aim to increase psychosocial health and wellbeing in the workplace. As argued in [14], “… good health is good work and there is growing evidence to support the case that workplace well-being interventions make good business sense” (p. 523). The importance of early and successful intervention, based on valid results, is receiving growing attention from organizations, as it is recognized as a cost-effective strategy.

A couple of potential limitations should be discussed. The first limitation of the studies is the absence of randomly selected samples. Both the organizations and the respondents who participated were not randomly drawn. Consequently, the generalizability of the results could be questioned. However, the three samples used were fairly heterogeneous, including a wide range of organizations and sectors.

Secondly, the three studies relied solely on self-report measures. Studies that rely on self-reports as the only measure of organizational behaviors have come under attack for two primary reasons: (1) self-reports are prone to many kinds of response biases, and (2) inferences about correlational and causal relationships may be inflated by the problem of common method variance [39].

One of our objectives was to construct a comprehensive tool so that its length would not be a deterrent for practical applications in various workplaces and future research. It takes about 20 to 25 minutes to complete the PRIWA, which can be completed group-wise (preferable, since the PRIWA was designed for group assessment, as part of a psychosocial risk evaluation) as well as individually. Seven mean scores are computed and reported, and the scales can be used independently, although its objectives are fully met when all psychosocial risks and outcomes are addressed.

An additional objective was that the tool should be applicable in all sectors of the labor market. A problem posed by generic questionnaires is that, sometimes, respondents may feel that they are asked to answer questions with little relevance to themselves. However, the main advantage of generic questionnaires is that they permit comparisons with normative values—provided that such values exist. During the last three years, we have administered PRIWA to more than 10 Greek companies, some of which have many employees (>1000). Thus, a critical mass of quantitative data has begun to form, which can be used to shed light on questions such as “how is our company doing compared with other workplaces?” and to develop industry-specific and national benchmarks. These values will enable workplaces to choose between two levels of comparison: their own industry or the country. Additionally, in collaboration with a large Greek Employee Assistance Programs provider, we have begun the collection of qualitative data through interviews and focus groups with employees of the companies to which PRIWA is administered. This way, its results can be used to design and implement actions and interventions to manage and limit psychosocial risks. 

## 5. Conclusions

Based on the results of the studies reported in the present study, we may conclude that the PRIWA is a valid and reliable psychometric instrument. The final version of the tool contains 68 items to measure occupational psychosocial risks (organizational culture, work demands, job insecurity, and presenteeism) and outcomes (health and wellbeing, dysfunctional behaviors, and job satisfaction and engagement), so it is a rather short and practical instrument. The PRIWA can therefore be considered a solid tool for conducting research to aid organizations in the development of a wellbeing policy or prevention plan in order to meet statutory regulations.

## Figures and Tables

**Table 1 behavsci-13-00104-t001:** Composition of Samples 1, 2, 3, 4, 5, and 6 according to sociodemographic characteristics.

Socio-Demographic Characteristics	Sample 1 f rf	Sample 2 * f rf	Sample 3 f rf	Sample 4 f rf	Sample 5 f rf	Sample 6 f rf
Total	1051 100	1309 100	494 100	402 100	283 100	149 100
Nationality Greek Other Missing	878 83.5	1182 90.3	469 94.9	402 100	261 92.2	137 91.9
	6 0.6	10 0.8	6 1.0		2 0.7	3 2.0
	167 15.9	117 8.9	10 2.0		20 7.1	9 6.0
Gender Male Female Missing	406 38.6 626 59.6 19 1.8	523 40.0 776 59.3 10 0.7	209 42.3 275 55.7 10 2.0	362 90.0 40 10.0	95 33.6 177 62.5 11 3.9	61 40.9 87 58.4 1 0.7
Age <25 years old 25-34 years old 35-44 years old 45-54 years old >55 years old Missing	149 14.2 402 38.2 282 26.8 134 12.7 43 4.1 41 3.9	143 10.9 476 36.4 383 29.3 216 16.5 60 4.6 31 2.4	23 4.7 147 29.8 151 30.6 125 25.3 30 6.1 19 3.6	17 4.2 217 54.0 150 37.3 18 4.5	14 4.9 76 26.9 69 24.4 83 29.3 22 7.8 19 6.7	24 16.1 40 26.8 33 22.1 39 26.2 13 8.7
Education Compulsory Lyceum University Graduate Missing	15 1.4 210 20.0 542 51.6 262 24.9 22 2.1	18 1.4 285 21.8 658 50.3 336 25.7 12 0.8	12 2.4 140 28.3 226 45.7 106 21.5 10 2.0	18 4.5 312 77.6 54 13.4 18 4.5	9 3.2 55 19.4 135 47.7 73 25.8 11 3.9	6 4.0 23 15.4 76 51.0 43 28.9 1 0.7
Marital status Single Married Living with partner Divorced Widowed Missing	557 53.0 353 33.6 61 5.8 52 4.9 4 0.4 24 2.3		190 38.5 250 50.6 9 1.8 33 6.7 3 0.6 9 1.8	49 12.2 309 76.9 11 2.7 33 8.2	94 33.2 140 49.5 18 6.4 18 6.4 2 0.7 11 3.9	70 47.0 63 42.3 8 5.4 6 4.0 2 1.3
Kids Yes No Missing	595 56.6 282 26.8 174 16.6		237 48.0 242 49 15 3.0	320 79.6 82 20.4	139 49.1 133 47.0 11 3.9	67 45.0 73 49.0 9 6.0
Employment sector Public Private Missing	325 30.9 569 54.2 157 14.9		76 15.4 406 82.2 12 2.4	402 100	169 59.7 113 39.9 1 0.4	62 41.6 82 55.0 5 3.3
Occupational status Higher management Lower management White-collar worker Blue-collar worker Other Missing	38 3.6 339 32.3 421 40.1 177 16.8 25 2.4 51 4.9		19 3.8 204 41.3 172 34.8 74 15.0 12 2.4 13 2.6	64 15.9 304 75.7 34 8.4	4 1.4 14 4.9 58 20.5 28 9.9 6 2.1 173 61.1	12 8.1 40 26.8 71 47.7 13 8.7 10 6.7
Job contract Permanent Temporary Self-employed Other Missing	699 66.5 107 10.2 75 7.1 8 0.8 162 15.4		395 80.0 45 9.1 42 8.5 12 2.4	402 100	210 74.2 47 16.6 14 4.9 1 0.4 11 3.9	99 66.4 20 13.4 22 14.8 2 1.3 6 4.0

* Only nationality, gender, age, and educational level were measured for Sample 2.

**Table 2 behavsci-13-00104-t002:** Retained items of the Psychosocial Risks and Impacts in the Workplace Assessment Tool and their corresponding standardized coefficients, pattern, and structure (Sample 1, N = 1051).

	Pattern Matrix	Structure Matrix
Factor 1—Organizational Culture (variance explained: 20.1%)		
My company fails to appreciate any efforts I make	0.72	0.77
My employer does not treat me fairly	0.72	0.74
In general, I feel that there is open and trustful communication between management and employees	−0.72	−0.72
I cannot count on my supervisor’s support	0.68	0.72
Even if I was the best employee at work, my company would fail to recognize it	0.66	0.72
My supervisor helps me and provides me with guidance, when facing problems at work	−0.65	−0.63
I believe that my supervisor is not aware of the problems I face at work	0.63	0.62
If it wasn`t for a few particular people at work, I would enjoy my job more	0.62	0.60
The work environment at the company I work for is sometimes hostile	0.60	0.64
My supervisor and the management do not keep me informed regarding whatever happens at work	0.60	0.65
The organization I work for cares more about the profit I can bring, rather than about me	0.54	0.65
The objectives and vision of the organization I work for, are not fully clarified	0.52	0.58
My role and responsibilities at work have not been fully clarified	0.51	0.58
At work, I have often been made to feel bad for the things that I do or say	0.43	0.55
Factor 2—Health & Well-Being (variance explained: 7.5%)		
I often get headaches	0.73	0.68
I often experience muscle pains	0.69	0.68
I often take painkillers	0.67	0.63
I often feel like I will collapse	0.64	0.74
I sometimes have a sudden feeling of numbness and tingling	0.61	0.61
I often feel physically exhausted	0.59	0.67
I often experience shortness of breath	0.59	0.66
Often my heart beats so fast that I cannot cool down	0.59	0.62
I frequently have fainting spells or I feel like it	0.57	0.63
Sometimes I feel afraid for no reason	0.52	0.61
Lately, I have had sleeping problems	0.50	0.58
I sometimes feel my arms and legs shaky	0.49	0.57
I find it difficult to relax	0.47	0.58
Lately, I get more emotional than in the past	0.43	0.45
Factor 3—Dysfunctional Behaviors (variance explained: 5.8%)		
I have been in trouble at work as a result of alcohol abuse	−0.78	−0.78
I sometimes use drugs or alcohol before and/or during work hours	−0.74	−0.74
I feel guilty about the amount of alcohol I consume	−0.68	−0.65
I drink alcohol in order to fall asleep	−0.67	−0.67
My use of substances is affecting my work performance	−0.61	−0.63
I often become the target of suggestive remarks at work	−0.58	−0.62
I have been threatened with physical assault at work	−0.52	−0.52
I cannot fall asleep if I don’t take sleeping pills	−0.49	−0.55
I have been sexually harassed at work	−0.42	−0.46
Factor 4—Job Satisfaction & Engagement (variance explained: 3.5%)		
I do not enjoy my job	−0.86	−0.87
I am proud of the work that I do	0.75	0.72
I don’t get any satisfaction from my job	−0.72	−0.78
My job is really a strain on me	−0.71	−0.80
The subject matter of my work does not require creativity	−0.69	−0.65
I remain at this work only due to financial reasons	−0.67	−0.74
I feel detached from my job	−0.66	−0.74
When I get up in the morning, I feel like going to work	0.64	0.64
Actually, I don’t really care about what happens at work	−0.62	−0.70
I regret my decision to follow this career path	−0.58	−0.64
I am not ambitious regarding my present job	−0.58	−0.64
I often feel like I am doing my job mechanically, without caring about the results	−0.55	−0.58
I can make decisions concerning my work	−0.40	0.37
Factor 5—Work Demands (variance explained: 3.0%)		
Due to my work, I don’t spend enough time with my family	0.68	0.69
My work is so demanding that there is no time left for a social life	0.67	0.70
I have no free time for myself	0.66	0.67
Due to the heavy workload, it is often expected of me to work overtime	0.66	0.66
I cannot find a healthy balance between my job and free time	0.64	0.68
I often have to work at a fast pace in order to complete my work	0.60	0.58
My job is constantly under demanding deadlines	0.58	0.59
I often feel that I have to leave everything behind for my job	0.58	0.58
Factor 6—Job insecurity (variance explained: 4.8%)		
I am worried about getting fired	0.82	0.83
I am afraid I will lose my job due to the financial crisis	0.79	0.81
I feel that, in the event I get fired, it will be difficult to find another job	0.61	0.62
The fact that my work is not permanent makes me feel insecure	0.59	0.60
I am concerned that I might be replaced by a better qualified employee	0.59	0.59
The potential for an eventual organizational change makes me feel insecure	0.56	0.61
I am worried that, due to organizational changes, I will be transferred to another position against my will	0.44	0.48
Factor 7—Presenteeism (variance explained: 2.7%)		
I am sometimes unable to work despite how hard I try	0.72	0.74
There are times when I am unable to accomplish anything at work because I don’t feel well	0.71	0.77
Usually, when something concerns me that is unrelated to my work duties, I am unable to work	0.68	0.69

**Table 3 behavsci-13-00104-t003:** Correlation matrix among the seven factors (Sample 1, N = 1051).

	F1	F2	F3	F4	F5	F6	F7
Factor 1	-	0.34 ***	0.21 ***	0.57 ***	0.26 ***	0.30 ***	0.31 ***
Factor 2		-	0.34 ***	0.30 ***	0.34 ***	0.37 ***	0.34 ***
Factor 3			-	0.28 ***	0.11 **	0.07 *	0.21 ***
Factor 4				-	0.04	0.30 ***	0.33 ***
Factor 5					-	0.12 ***	0.14 ***
Factor 6						-	0.21 ***
Factor 7							-

* *p* < 0.05; ** *p* < 0.01; *** *p* < 0.001.

**Table 4 behavsci-13-00104-t004:** Summary of Fit Indices for the PRIWA versions (Sample 2, N = 1309).

Model	χ^2^/df	CFI	TLI	SRMR	RMSEA	RMSEA CI 90%	AIC
M_1_	90.87	0.47	0.46	0.100	0.082	0.081–0.083	22,082.7
M_2_	30.28	0.87	0.86	0.057	0.042	0.041–0.043	7490.8
M_3_	20.69	0.93	0.92	0.055	0.036	0.035–0.037	6195.8

*Note*: M_1_: Single-factor model for PRIWA; M_2_: Seven-factor model for PRIWA; M_3_: Seven-factor model for PRIWA with covariances among errors.

**Table 5 behavsci-13-00104-t005:** Descriptive statistics and internal consistency coefficients (Cronbach’s alpha) for the PRIWA scales (Sample 3, N = 494).

	Items	Mean	SD	Cronbach’s α
Factor 1—Organizational Culture	14	40.7	11.3	0.91
Factor 2—Health & Well-Being	14	33.0	10.4	0.89
Factor 3—Dysfunctional Behaviors	9	11.5	3.2	0.74
Factor 4—Job Satisfaction & Engagement	13	28.1	9.7	0.92
Factor 5—Work Demands	8	24.2	6.1	0.82
Factor 6—Job insecurity	7	20.9	5.4	0.80
Factor 7—Presenteeism	3	7.6	2.7	0.74

**Table 6 behavsci-13-00104-t006:** Pearson’s Correlation Coefficients between PRIWA scales and COPSOQ II, EAS, and SPS-6 measures.

PRIWA F1 × COPSOQ II (Predictability, Quality of leadership, Social support, Feedback at work, and Sense of community) ^1^	r(402) = 0.76 ***
PRIWA F2 × COPSOQ II (General Health, Mental Health, and Vitality) ^2^	r(472) = 0.76 ***
PRIWA F3 × COPSOQ II (Behavioral, Somatic & Cognitive Stress) ^2^	r(478) = 0.64 ***
PRIWA F4 × UWES (Vigor) ^3^	r(283) = 0.70 ***
PRIWA F4 × UWES (Dedication) ^3^	r(283) = 0.83 ***
PRIWA F4 × UWES (Absorption) ^3^	r(283) = 0.60 ***
PRIWA F5 × COPSOQ II (Demands) ^2^	r(476) = 0.74 ***
PRIWA F6 × COPSOQ II (Insecurity at work) ^2^	r(475) = 0.67 ***
PRIWA F7 × SPS-6 ^4^	r(145) = 0.62 ***

*Notes*: ^1^ Sample 4 (N = 402); ^2^ Sample 3 (N = 494); ^3^ Sample 5 (N=283); ^4^ Sample 6 (N = 149). PRIWA F1: Factor 1—Organizational Culture; PRIWA F2: Factor 2—Health & Well-Being; PRIWA F3: Factor 3—Dysfunctional Behaviors; PRIWA F4: Factor 4—Job Satisfaction & Engagement; PRIWA F5: Factor 5—Work Demands; PRIWA F6: Factor 6—Job insecurity; PRIWA F7: Factor 7—Presenteeism; COPSOQ II: Copenhagen Psychosocial Questionnaire; UWES: Utrecht Work Engagement Scale; SPS-6: Stanford Presenteeism Scale. *** *p* < 0.001.

## Data Availability

The data presented in this paper are available on request from the corresponding author. The data are not publicly available due to privacy restrictions.

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
