# Peer review of "The Psychosocial Risks and Impacts in the Workplace Assessment Tool: Construction and Psychometric Evaluation"

_behavsci, 2023, doi:10.3390/bs13020104_

Round 1

Reviewer 1 Report

This is a well written and useful paper. This paper describes an evaluation tool and due to the way the sampling of participants was done there are limitations which is also described.

It would be good to strengthen the discussion section by explaining how this methodology can be applied in the field and how at a next stage it can be adopted to have less of the limitations described.

Author Response

Thanks for your review and your comment, which both reviewers made. I have added lines 436-440 and 442-446 to give the reader an idea about what we have done and are currently doing on this issue.

Reviewer 2 Report

Manuscript ID: behavsci-2093939

Title: The Psychosocial Risks and Impacts in the Workplace Assessment Tool: Construction and Psychometric Evaluation

Thank you for providing a chance to review this manuscript.

Comment: Major revision.

Detailed information:

Keywords

Line 21, page 1: The order of the keywords should be adjusted, and I think it is reasonable to advance "Psychometric tool" and "PRIWA" in the context of the title and main content of the article.

Introduction

Lines 24-27, page 1: “Work practices, processes and conditions are constantly changing ... employment trends etc.” I think these two sentences are similar in meaning and seem to be repetitive, and they do not lead well to the later "Psychosocial risks", so I suggest revising them.

In this section you spend too much space on "Psychosocial risks", but I am more interested in the development of "The Psychosocial Risks and Impacts in the Workplace Assessment Tool (PRIWA)" and how it has been validated in different countries and regions. You should clarify the focus of your writing. This section is so long that if I were the reader I would have a hard time having the patience to read it all, please try to streamline it.

Materials and Methods

Lines 167-168, page 4: I don't think a sentence is necessary as a separate paragraph, please combine it.

Lines 184-216, page 2: I would like to know the psychometric properties of these three scales, please add.

Table 1

A standard three-line table should be used.

Discussion

 I'm glad to see you mentioned the strengths and weaknesses of the study at the moment, and would also like to know more about what the future outlook of this study is?

Thank you and my best,

Your reviewer

Author Response

Thank you very much for your valuable comments. My responses appear in red under each comment.

Line 21, page 1: The order of the keywords should be adjusted, and I think it is reasonable to advance "Psychometric tool" and "PRIWA" in the context of the title and main content of the article.

Done!

Lines 24-27, page 1: “Work practices, processes and conditions are constantly changing ... employment trends etc.” I think these two sentences are similar in meaning and seem to be repetitive, and they do not lead well to the later "Psychosocial risks", so I suggest revising them.

Thanks for the comment; you are right. The first sentence was deleted.

In this section you spend too much space on "Psychosocial risks", but I am more interested in the development of "The Psychosocial Risks and Impacts in the Workplace Assessment Tool (PRIWA)" and how it has been validated in different countries and regions. You should clarify the focus of your writing. This section is so long that if I were the reader I would have a hard time having the patience to read it all, please try to streamline it.

I have considered this comment very carefully, but I decided to leave it for the following reasons: The first and most important is that a theoretical background was necessary to explain why the PRIWA addressed those dimensions of psychosocial risks. The second reason is that in a previous submission to a psychometrics journal, the reviewers stressed the importance of a solid theoretical background and made many comments that have led to its present form. Finally, the second reviewer was pleased with the introduction.

Lines 167-168, page 4: I don't think a sentence is necessary as a separate paragraph, please combine it.

I imagine that you mean lines 176-178. Done!

Lines 184-216, page 2: I would like to know the psychometric properties of these three scales, please add.

The following lines have been added: 193-195, 202-206 and 214-216.

Table 1

A standard three-line table should be used.

Although I think the lines help to make the table readable, I have followed your suggestion. 

Discussion

I'm glad to see you mentioned the strengths and weaknesses of the study at the moment, and would also like to know more about what the future outlook of this study is?

This was a comment made by both reviewers. I have added lines 436-440 and 442-446 to give the reader an idea about what we have done and are currently doing on this issue.

Round 2

Reviewer 2 Report

Manuscript ID: behavsci-2093939

Title: The Psychosocial Risks and Impacts in the Workplace Assessment Tool: Construction and Psychometric Evaluation

I am happy to see the improvement of your manuscript, but there are still the following problems.

Comment: Minor revision.

Detailed information:

Introduction

Lines 70-73, page 2: Are these data you mentioned supported by the literature, and if so, please add it.

Materials and Methods

Line 199, page 4: Has the Greek version been validated in the population and what are the results?

Lines 264-266, page 7: What are the inclusion and exclusion criteria for the sample? What was the quality control of the paper-and-pencil questionnaires survey and the online survey conducted?

Results

Lines 351-353, page 10: You mentioned " The results confirmed the good fit between the model and the observed data ", Please point out the references that indicate these values are good.

Thank you and my best,

Your reviewer

Author Response

Many thanks for your comments. My responses appear under your comments and suggestions:

Introduction

Lines 70-73, page 2: Are these data you mentioned supported by the literature, and if so, please add it.

In lines 70-73, there are two arguments presented:

The first one is about the underestimation (from employers) of the extent to which employees are experiencing mental illness. One reference [14] (Cooper & Dewe, 2008) was included to support this; a second one and more recent publication (Shaw Trust, 2010) was added:

Shaw Trust. Mental Health: Still The Last Workplace Taboo? London, UK, 2010. Available online: https://trajectorypartnership.com/wp-content/uploads/2013/09/Mental-Health-Report.pdf (accessed on 23 January 2023).

The second argument is about how common is absence from one’s work due to mental disorders and suggests that it can be compared to absence due to musculoskeletal diseases. The following reference was added to support it:

Kausto, J.; Pentti, J.; Oksanen, T.; Virta, L.J.; Virtanen, M.; Kivimäki, M.; Vahtera, J. Length of sickness absence and sustained return-to-work in mental disorders and musculoskeletal diseases: a cohort study of public sector employees. Scand J Work Environ Health 2017, 43, 358–366. DOI: 10.5271/sjweh.3643

Materials and Methods

Line 199, page 4: Has the Greek version been validated in the population and what are the results?

We have used the Greek version of the tool, which has been used by Schaufeli et al. (2006) in their cross-national study (this is reference [26] in the paper). The factorial validity of the tool has been demonstrated through CFA (in Greece and many other countries) and was reported in the paper. I am unaware of any other validation effort; therefore, no changes have been made to this part.

Lines 264-266, page 7: What are the inclusion and exclusion criteria for the sample? What was the quality control of the paper-and-pencil questionnaires survey and the online survey conducted?

Inclusion criteria were employees aged 18 years old and above who were proficient in the Greek language and agreed to participate in the study. Exclusion criteria were employees who did not give their consent and were not proficient in the Greek language.

Those two sentences now appear in lines 268-270, page 7.

As far as quality control is concerned, all questionnaires (paper-and-pencil and online) were administered to employees only through the HR department of their companies. Furthermore, two quality checks have been incorporated into the survey: 1) a straight-lining flag was used in both offline and online versions, and 2) a speeder check was used in SurveyMonkey to flag participants who finished the survey in a brief period. However, the information was included in the paper simply because the paper is already very long, and usually, this info does not appear in similar papers. However, if you insist that this should be included, I could add a few lines.

Results

Lines 351-353, page 10: You mentioned "The results confirmed the good fit between the model and the observed data ", Please point out the references that indicate these values are good.

The following reference was added to the ones already included in the paper (see lines 340 and 351):

Marsh, H.W.; Hau, K.-T.; Grayson, D. Goodness of Fit in Structural Equation Models. In Contemporary psychometrics: A festschrift for Roderick P. McDonald; Maydeu-Olivares, A., McArdle, J.J., Eds.; Lawrence Erlbaum Associates Publishers, 2005; pp. 275–340.